# Preparation and Tribological Properties of Bismaleimide Matrix Composites Reinforced with Covalent Organic Framework Coated Graphene Nanosheets

**DOI:** 10.3390/polym14163289

**Published:** 2022-08-12

**Authors:** Chao Liu, Xin Xue, Qiming Yuan, Yang Lin, Yan Bao, Yinkun He, Wenbo Zhang

**Affiliations:** 1Key Laboratory of Auxiliary Chemistry and Technology for Chemical Industry, Ministry of Education, Shaanxi Collaborative Innovation Center of Industrial Auxiliary Chemistry and Technology, Shaanxi University of Science and Technology, Xi’an 710021, China; 2College of Bioresources Chemical and Materials Engineering, Shaanxi University of Science and Technology, Xi’an 710021, China; 3College of Chemistry and Chemical Engineering, Shaanxi University of Science and Technology, Xi’an 710021, China

**Keywords:** COFs, mechanochemical synthesis, graphene, tribological

## Abstract

The poor compatibility between the polymer matrix and complex modification processes greatly affects the excellent tribological properties of graphene in the polymer matrix. In this study, a covalent organic framework (COF)-coated graphene hybrid lubricating filler (G/COFs) was synthesized in situ using a sample one-step mechanochemical synthesis process. This was used to improve the tribological properties of bismaleimide (BMI) resin. The morphology and microstructure of the G/COFs hybrid were characterized, and the effect of the added amount on the tribological properties of the G/COFs/BMI composites was studied. The results showed that the G/COFs hybrid could improve the stability of the friction coefficient and decrease the volume wear rate of BMI composites. Compared to the neat BMI, the 0.6 wt% G/COFs/BMI composites showed optimal tribological performance, with the friction coefficient and volume wear rate decreasing from 0.35 to 0.14 and from 48 × 10^−6^ to 10.6 × 10^−6^ mm^3^/(N‧m), respectively. In addition, the G/COFs/BMI composites showed lower friction coefficient fluctuations and volume wear rates than G/BMI composites. This is mainly attributed to the fact that the deposition of COFs can not only effectively prevent the aggregation of graphene nanosheets, but can also significantly improve the compatibility and interfacial bond between the graphene and BMI matrix. Moreover, the good synergistic effect between the lamellar COFs and graphene nanosheets can generate high-quality self-lubricating transfer films during the friction process. The excellent dispersibility, efficient chemical functionalization, better friction reduction and wear-resistance properties, and facile preparation method make graphene/COFs hybrid nanoparticles promising as an excellent lubricating filler.

## 1. Introduction

Due to their significant mechanical strength and high-temperature resistance, polymer materials are widely applied in aerospace, army facilities, healthcare equipment, automatic processing, and other fields [1]. Zhou et al. [2] designed and synthesized a novel eugenol-functionalized cage-like polyhedral oligomeric silsesquioxane (EG-POSS) that exhibits excellent solubility and reactivity with BD-type bismaleimide resin. Thus, when 4 wt% EG-POSS is incorporated and uniformly dispersed in the resin matrix, the resultant hybrid (BDEP-0.04) shows outstanding comprehensive performance, especially in terms of dielectric, mechanical, and thermal properties. However, polymeric materials exhibit poor self-lubricating properties due to their terrible operating environments, reducing the life of these materials [3,4]. It has been demonstrated by many researchers that the employment of self-lubricating admixtures as lubricants in polymer matrix composites improves the lubricating environment, thus reducing frictional force and improving wear life. Some typical two-dimensional nanoparticles such as graphene, molybdenum disulfide, MXene, etc., exhibit excellent friction reduction and wear resistance [5,6,7].

Graphene (G) has been widely introduced into the polymer matrix as a lubricant nanofiller to improve friction reduction and the wear-resistance of composites due to its high thermal conductivity, significant mechanical properties, and superior friction and abrasion properties [5,8]. However, the π-π bonding interaction between graphene nanosheets may result in the agglomeration and sinkage of graphene in the polymer matrix, which significantly limits its strengthening and lubricating performance [9,10]. Researchers have found that modifying graphene is regarded as an effective method to solve its dispersion in the polymer matrix [11]. To date, the most common modification strategies for this aim have involved covalent and non-covalent modifications [12,13,14]. To improve graphene dispersion in bismaleimide (BMI) resin and the interfacial interaction between them, our group has reported on polytriazine surface-modified graphene. We found significant improvements in the compatibility and interfacial strength between graphene and BMI substrates through effective surface modifications [15]. Although the covalent modifications solve the problem of poor dispersion in the matrix, the conjugation of the graphene nanosheets was destroyed, reducing its tribological properties. In contrast to covalent modification, the advantage of non-covalent modification is that it improves dispersion without destroying the structure or the excellent properties of graphene or graphene oxide [12,16,17]. Nobile et al. [17] prepared modified graphene (G-py) based on a non-covalent π-π reaction between graphene (G) and a pyrene-based molecule (py). Compared to the unmodified graphene, the G-py showed good dispersibility in epoxy resin.

Covalent organic frameworks (COFs) [18,19,20,21] have attracted extensive attention due to their advantages of high surface area, adjustable structure, abundant active sites, and excellent crystallinity. In particular, the abundance of active sites conducive to modification and that have an interatomic layer with weak shear strength makes COFs an ideal choice for adding lubricants to reduce friction and wear [22,23,24,25]. Zhang et al. [26] obtained DDP@TD-COF via an amine–aldehyde condensation reaction of the triazine compound and vinyl-functionalized monomers through a solvothermal process to form a vinyl-functionalized COF (TD-COF). This was followed by the covalent bonding of commercial lubricating molecules (DDP) via the UV-induced thiol-ene “click” reaction. By adding 0.05 wt% of the DDP@TD-COF to the 500SN base oil, the compound lubricating oil showed excellent friction reduction and anti-wear ability. However, the process of preparing COFs using a solvothermal process is harsh (120 °C, 72 h). Mechanical synthesis has the advantage of simple operation and solves the complex problem of its preparation technology [27,28,29]. However, COF-based composites have been widely reported. As far as we know, graphene/COFs are mostly used in adsorption [30,31], oil/water separation [32], and energy storage [33]; they are rarely reported in the field of friction.

In the present work, with graphene oxide, p-Phenylenediamine, and 1,3,5-Benzenetricarbonyl trichloride as raw materials, a composite lubricating filler (G/COFs) was synthesized via a one-step mechanochemical synthesis. Following, the G/COFs hybrid was introduced into bismaleimide (BMI) resin and G/COFs/BMI composites were cast. The morphology and microstructure of the G/COFs were characterized by FT-IR, XRD, and TEM, and the effects of the added amount on the tribological properties of the BMI composite were studied. The results revealed that the tribological properties of the BMI composite were significantly improved by introducing the G/COFs hybrid. The use of COF-modified G not only avoids graphene nanosheet agglomeration, but the lamellar COFs and graphene nanosheets also have a good synergistic effect, resulting in the composites being able to generate high-quality self-lubricating transfer films during the friction process.

## 2. Materials and Methods

### 2.1. Materials

Concentrated sulfuric acid (H_2_SO_4_, 98%) and concentrated hydrochloric acid (HCl, 98%) were obtained from Sinopharm Chemical Reagent Co., Ltd. (Shanghai, China). Potassium persulfate (K_2_S_2_O_8_, 99.5%), phosphorus pentoxide (P_2_O_5_, 98%), and anhydrous ethanol (99.7%) were sourced from Tianjin Fuyu Fine Chemical Co (Tianjin, China). Natural graphite flakes (325 mesh) were obtained from Qingdao Hensen Graphite Co., Ltd. (Qingdao, China). Sodium hydroxide (NaOH, 95%), p-Phenylenediamine (97%), and 1,3,5-Benzenetricarbonyl trichloride (98%) were obtained from Shanghai Macklin Biochemical Co. (Shanghai, China). 4,4′-Bismaleimidodiphenylmethane (BDM, industrial grade) and diallyl bisphenol A (DBA, industrial grade) were purchased from Wuhan Zhisheng Technology Co. (Wuhan, China). All reagents were applied without further purification.

### 2.2. Preparation of Graphene Oxide

K_2_S_2_O_8_ (5 g) and P_2_O_5_ (5 g) were mixed and dissolved in H_2_SO_4_; then, the graphite was added to an oil bath at 80 °C and stirred and kept for 6 h. After cooling to room temperature, the mixture was rinsed with deionized water to neutralize it, and the powder was dried at 60 °C for 12 h to obtain a black powder. 

### 2.3. Preparation of G/COFs

Firstly, graphene oxide (100 mg), p-Phenylenediamine (140 mg), and 1,3,5-Benzenetricarbonyl trichloride (160 mg) were added to a grinding pot with a certain number of agate balls and ground in a low-temperature planetary ball mill at 500 r/min for 90 min. Next, the pot was filled with 80 mL of 5% aqueous sodium hydroxide solution, and milling was maintained for 30 min. Finally, the product was filtered through a filter membrane and washed with deionized solution and ethanol, and it was finally dried under vacuum at 70 °C for 12 h. The black powder obtained was the prepared G/COFs.

### 2.4. Preparation of G/COFs/BMI

The G/COFs/BMI composites were prepared by mixing pre-weighed quantities of G/COFs, DBA, and BDM (DBA and BDM with a mass ratio of 3:4). Then, the mixture was heated to 135 °C until the mixture had totally melted and the G/COFs had uniformly dispersed. Finally, the mixture was cured following the schedule of 150 °C/2 h + 180 °C/2 h + 220 °C/4 h. The post-curing process was 250 °C/4 h. The composites based on BMI, including neat BMI and G/BMI composites with different filler contents [16,34], were obtained following the methods described in our previous work. The fabrication process of the G/COFs hybrid and its BMI composite is presented in Figure 1.

### 2.5. Characterization

The Fourier transform infrared (FT-IR, VECTOR-22, Bruker, Karlsruhe, Germany) spectra of samples were characterized. The sample phases were determined by an X-ray diffractometer (XRD, Smart Lab 9Kw, Rigaku, Tokyo, Japan). The sample morphology was recorded by a transmission electron microscope (TEM, G2 TF20 S-TWIN, FEI, Hillsboro, OR, USA). A small amount of the sample was dispersed into ethanol and sonicated to make it uniformly dispersed. The sample was dropped onto a 230-mesh carbon support film to obtain a transmission electron microscope sample. The surface morphology of the samples was observed using a scanning electron microscope (SEM, S-4800, Hitachi, Tokyo, Japan). The tribological properties of the samples were examined using a tribological test machine (MMUD-1B, Jinan Hengxu Testing Machine Technology Co., Ltd., Jinan, China). The experiments were conducted with end-face friction sub-materials made of 45# steel, and friction and abrasion tests were performed on a testing machine under dry conditions. The hardness properties of the composite materials were tested by a microhardness tester (Micro Vicker HV30, Maga Instruments (Suzhou) Co., Ltd., Suzhou, China). The thermal properties of the composites were analyzed using a thermogravimetric analyzer (TGA, Q500, TA, New Castle, DE, USA) in a N_2_ atmosphere at a heating rate of 10 °C/min. 

## 3. Results and Discussion

### 3.1. Structural Analysis of G/COFs

FT-IR was used to study the chemical structures of p-Phenylenediamine, 1,3,5-Benzenetricarbonyl trichloride, graphene oxide, COFs, and G/COFs. As shown in Figure 1, 1,3,5-Benzenetricarbonyl trichloride exhibits two clear absorption peaks at 1752 and 704 cm^−1^, which belong to the C=O stretching vibrations and C−Cl stretching vibrations in the acyl chloride groups, respectively [29]. In the spectrum of p-Phenylenediamine, the peaks at 3380 and 3302 cm^−1^ belong to the anti-symmetric and symmetric stretching vibration peaks of −NH_2_, respectively. In the spectrum of the COFs, the peaks at 3051 cm^−1^ and 1514 cm^−1^ belong to the C−H and C=C stretching vibrations of the benzene ring, respectively. The peaks at 824 cm^−1^ and 708 cm^−1^ belong to the characteristic substitution peaks on the benzene rings [27,28]. In addition, it can be noted that there was a disappearance of the −NH_2_ stretching band (3202 and 3380 cm^−1^) in p-Phenylenediamine and the C−Cl stretching (704 cm^−1^) in 1,3,5-Benzenetricarbonyl trichloride. Meanwhile, two new peaks occurred at 1253 cm^−1^ and 1660 cm^−1^, which were assigned to the absorption III (C=N) and Ⅰ (C=O) bands of −CONH− [35,36]. These measurements indicate that the COFs were successfully prepared. Graphene oxide exhibits three clear absorption peaks at 1062, 1315, and 3600 cm^−1^, which belong to the C−O−C stretching vibrations, C−H bending vibration, and −OH stretching vibrations, respectively [16,34]. The FT-IR peak positions of the G/COFs are almost identical to those of the COFs, which indicates that the loading of G has less influence on the structure of the COFs. 

XRD analysis was used to study the structure of the sample. Figure 2 describes the XRD patterns of the graphene oxide, G, COFs, and G/COFs. The XRD pattern of graphene oxide shows a wrapper peak at 11.3°, corresponding to the (001) crystal surface of graphene oxide [37,38]. Compared to graphene oxide, the XRD spectrum of G shows a wrapper peak at 24.3°, which corresponds to the (002) crystal plane of graphene. This shows that the functional groups on the surface of graphene oxide were removed and that the graphene was successfully prepared [39]. For the XRD pattern of the COFs, the strongest diffraction peak at 2θ = 26.2° corresponds to the (001) crystallographic surface of the COFs [27,29]. As the COFs are coated to the graphene surface, the characteristic diffraction peak at 26.2° becomes weaker, as evidenced by the XRD spectra of G/COFs [30,32]. This may be because the COFs growth on the surface of the graphene can restrict the number of graphene layers by preventing the stacking of graphene sheets to some extent. Because graphene’s layer-to-layer interactions are weak, the distinctive diffraction peak weakens.

### 3.2. Microstructure of G/COFs Hybrid

To identify the successful fabrication of COFs and G/COFs, the morphological structure of the samples was analyzed by TEM. As shown in Figure 3a, the G surface shows many wrinkles with a typical peeling muslin morphology [16,34,40,41]. Figure 3b shows that COFs have lamellar structures [28,42,43,44]. Compared to the COFs, the G/COFs show a wrinkled lamellar structure, which is mainly due to the difference in morphology under shear force during graphene ball milling, and the change further confirms the growth of COFs on the surface of G. These layers are composed of numerous layers of nanoscale lamellar planes, allowing the relative sliding between layers to be easier to achieve under frictional shear stresses (Figure 3a–c). The energy dispersive spectrometer mapping of the G/COFs (Figure 3e–g) shows a uniform distribution of the elements C, N, and O, indicating that the COFs molecules have successfully coated on G, with coating being uniformly distributed along the organic carbon backbone.

### 3.3. Tribological Properties of G/COFs/BMI Composites

The friction coefficient and volume wear rate of composites with various filler contents are shown in Figure 4a. As shown, the friction coefficient and volume wear rate of the composites fall at first and then rise as the number of fillers increases. When the filler addition is 0.6 wt%, the friction and volume wear rate coefficients of the G/COFs/BMI composites reach their lowest values of 0.14 and 10.6 × 10^−6^ mm^3^/(N·m). Compared to neat BMI (0.35 and 48 × 10^−6^ mm^3^/(N·m)), they decreased by 60% and 77.9%, respectively. The reduction of the friction coefficient of the G/COFs compared to BMI is significantly better than similar reductions seen in the literature [16,45]. However, when the addition amount is more than 0.6 wt%, the friction coefficient and volume wear rate of G/COFs/BMI composites increase. However, they are still lower than those of neat BMI. The reason for this phenomenon is that the fillers occupy too large a proportion in the resin matrix, forming inhomogeneous aggregates that make it difficult to form a continuous and uniform self-lubricating transfer film on the surface of the metal counterpart ring [46].

To obtain the tribological mechanism of G/COFs, the tribological properties of neat BMI, 0.6 wt% G/BMI composites, and 0.6 wt% G/COF/BMI composites were studied, as shown in Figure 4b. After the addition of 0.6 wt% G into BMI, the friction coefficient was 0.27. Compared to the neat BMI, the friction coefficient could be reduced by 20.5%. The addition of the same amount of G/COFs resulted in a composite material with excellent stability and good friction reduction properties [34,47]. Meanwhile, the preparation process for G/COFs is straightforward and can be mass produced. Therefore, G/COFs hybrid nanoparticles are promising as a new lubricating filler and have been applied to the field of friction.

To further investigate the tribological suitability of G/COFs/BMI composites, the tribological properties of G/COFs/BMI composites under different loads and rotational speeds were studied at 0.6 wt% filling. Figure 5a shows the effect of the applied load on the friction coefficient and volume wear rate of 0.6 wt% G/COFs/BMI composites. It can be observed that the friction coefficient and volume wear rate of the composite decrease gradually as the load gradually increases from 50 N to 200 N. This is mainly because as the load increases, the contact area between the metal counterpart ring and the composite becomes more extensive, thus enabling the formation of a large self−lubricating transfer film in the friction contact area. This prevents direct exposure between the composite and metal counterpart ring. However, when the applied load force exceeds 200 N, the friction coefficient and volume wear rate of the composite material show an increasing trend. The main cause of this phenomenon is that the wear surface of the composite is cracks easily under high loading conditions, which leads to an increase in the number of fractured resin fragments [48].

Figure 5b shows the effect of rotational speed on the tribological properties of 0.6 wt% G/COFs/BMI composites under a load of 200 N. The friction coefficient and volume wear rate of the composites show a decreasing and then increasing trend as the rotational speed improves (See Figure 5b). This is because it is difficult for the composite material to form a self-lubricating transfer film on the metal counterpart ring at a low rotational speed. The rough metal counterpart ring directly wears against the composite material; thus, its tribological properties are poor [49]. When increasing the rotational speed, the self-lubricating transfer film gradually forms on the surface of the metal counterpart ring, and its friction coefficient is steadily reduced. When the rotational speed is 200 r/min, the friction coefficient and volume wear rate of the composite materials are reduced to their lowest values. However, when the speed of rotation exceeds 200 r/min, the frictional coefficient and volume wear rate of the composites material gradually increase. The main reason for this phenomenon is that the frictional heat accumulation leads to a sharp rise in the surface temperature of the composite material, and thus, adhesive wear occurs, increasing the frictional coefficient and volume wear rate of the composite [50].

### 3.4. Friction and Wear Mechanism Analysis

To study the wear mechanism of G/COFs/BMI composites, SEM was adopted to analyze the worn surface of the composites. Figure 6a shows that the worn surface of the neat BMI is very uneven, with plastic deformation and a large number of resin fragments. Its wear mechanism is mainly adhesive abrasion. The worn surface of the 0.6 wt% G/BMI composites is rougher than that of neat BMI, showing a scale-like morphology with broad and deep abrasion marks. This is a typical characteristic of the simultaneous existence of adhesive wear and abrasive particle wear. The worn surface without a large number of resin fragments indicates that the introduction of G can improve the load-bearing capacity of the BMI matrix. As shown in Figure 6c, the 0.6 wt% G/COFs/BMI composites show a smooth worn surface with shallow abrasion marks and small abrasive grains, which are typical abrasive grain wear characteristics. This is mainly explained by the fact that the introduction of G/COFs composite particles can form a dense and flat self-lubricating transfer film on the metal counterpart ring surface, avoiding direct contact between the rough metal counterpart ring and the BMI composite. These are consistent with the characterization tribological performance results [51,52]. The outstanding tribological properties of the G/COFs/BMI composites are mainly attributed to two aspects. Firstly, graphene has excellent reinforcement efficiency and self-lubricity, so the introduction of graphene into the polymer matrix can effectively improve the mechanical strength of the composite, thereby improving the wear resistance of the composite (See Appendix A). Secondly, the deposition of COFs can not only effectively prevent the aggregation of graphene nanosheets, but can also significantly improve the compatibility and interfacial bond between the graphene and polymer matrix. Therefore, the G/COFs/BMI can form a more uniform and continuous high-quality self-lubricating transfer film during the friction process to effectively improve the tribological properties of the composites.

### 3.5. Thermal Stability of G/COFs/BMI Composites

Because of the enormous amount of heat generated during the friction process, the thermal stability of the composites is critical for their tribological qualities. Figure 7 shows the TGA results for neat BMI, 0.6 wt% G/BMI, and 0.6 wt% G/COFs/BMI composites. Compared to neat BMI, the addition of G and G/COFs hybrid nanoparticles increases the thermal decomposition temperature of BMI in the primary stage. This is mainly attributable to the ability to prevent G stacking and to increase the volume share of G when COFs are loaded on the G surface. Large graphene nanosheets can form physical barriers to block the chain movement and delay the thermal decomposition of the resin matrix. It is suggested that the addition of G/COFs can improve the thermal stability of their compounds. Meanwhile, the introduction of fillers can significantly improve the final residual carbon yield of their composites because of the excellent catalytic carbon production ability of graphene [53].

### 3.6. Hardness Performance of G/COFs/BMI Composites

Hardness is a parameter that resists various permanent shape changes when subjected to forces, and the hardness of a material is strongly correlated with its frictional properties. The correlation between the hardness of G/COFs/BMI composites and the G/COFs content is shown in Figure 8. The hardness of G/COFs/BMI composites increases continuously with the addition of G/COFs, reaching a maximum value (45.6) at 0.6 wt% G/COFs. This indicates the filling with G/COFs can effectively improve the hardness of their composites. However, when the filler content is further increased, the hardness of the composites is reduced but is still better than that of neat BMI (34.8). This may be ascribed to the increase in the number of fillers, as more graphene accumulates in the resin, resulting in lower voids in the material, which reduces the hardness of the material [54]. 

## 4. Conclusions

In summary, novel G/COFs hybrid nanoparticles were synthesized in one step via mechanochemical synthesis. This method not only avoids the use of harmful solvents, but can also realize the scaled-up production of composite nanomaterials. The G/COFs/BMI composites show superior tribological characteristics compared to neat BMI and G /BMI composites. When the G/COFs addition is merely 0.6 wt%, the frictional coefficient and volume wear rate of their BMI composites are decreased to 0.14 and 10.6 × 10^−6^ mm^3^/(N‧m). These are lowered by 60% and 77.9% compared to neat BMI, respectively. Even in harsh friction environments, the tribological characteristics of the G/COFs/BMI composites are still superior to those of neat BMI. This is mainly attributed to the fact that COF-coated graphene can not only effectively improve compatibility with the polymer matrix, but can also effectively play a role in friction reduction and wear resistance, thus synergistically improving the tribological properties of their composites. We believe that these novel G/COFs hybrid nanoparticles have the potential to become the lubricating filler for large-scale production applications, such as for the production of various automotive parts. However, this method is limited to G/COFs hybrid nanoparticles that need to be synthesized at a high temperature. 

## Data Availability

Not applicable.

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
