# Peer review of "Preparation and Tribological Properties of Bismaleimide Matrix Composites Reinforced with Covalent Organic Framework Coated Graphene Nanosheets"

_polymers, 2022, doi:10.3390/polym14163289_

Round 1

Reviewer 1 Report

General Comment: I want to congratulate the authors for the manuscript titled as “Preparation and tribological properties of bismaleimide matrix composites reinforced with covalent organic framework coated graphene nanosheets”. This is an interesting study and the manuscript in its present form is well formatted. I think the results and methods sections are well done with some minor corrections to be made.

Abstract: writing is too generalized. The main theme of this paper is not described in the abstract. Abstract section should be concisely reflected the content and summarize the problem, the method, the results, and the conclusions. Also, please add qualitative or quantitative results of your work. Besides, please avoid starting the sentence with conjunctions such as “and”.

The introduction section has been written beautifully but need to include recent published papers on bismaleimide-based composites and their production process and overall properties. In this section, more literature papers have to be included to explain the subject better. Some giving citations need to be check such as [7-9] [20-23], as they may not provide the required information in a sentence. In addition, your reference list contains a few papers from Polymers journal. If your work is convenient for this journal’s context, then several references should be included from Polymers journal.

(Please only refer to the most relevant and recent studies.)

At the last paragraph of the introduction, please clearly show the general outline of the paper and show the importance of the study along with the main aim. Please show the literature gaps demonstrating the presented study fills it.

There is an interesting approach and design exists, I just propose to emphasis the practical significance of the presented methodology in several points of article.

Language used in the manuscript is generally satisfying. However, writers should pay more attention of singular / plural nouns. Also, they should control the spell check/ punctuation of words and sentences. Please check all manuscript for typos and misspellings. Also, please recheck upper/lower case letter and text format for example in abstract line 17 “graphene” is bigger than other words. Please revise such inconsistences.

 In Fig. 6, some texts are not readable (especially scale bars). Please revise them.

The XRD diffraction pattern of Figure 2 should be marked with a standard PDF card. Also, the referred PDF card No. in XRD results should be supplemented in the corresponding text of Fig. 2.

To put the current work in a broader context, some high-quality literatures should be cited. From my point of view, the topic is surely interesting for the Readers of Polymers as well as the paper is quite well structured. Nevertheless, some results should be better explained avoiding too generic statements (without experimental evidence). In particular, the discussion of the FT-IR results must be checked and mandatory improved in order to increase the quality and readability of this paper. Results are explained without any proper references. Improve the results and discussion and conclusion parts. The results and discussion section should be widened with more focusing point of the findings. And these sentences should be supported with the literature studies. Results and discussion parts are inadequate according to citation and analyze in detail. There should be the importance of the study in detail, comparison results with other approaches in literature, the success of the prediction and computational results.

Furthermore, the Authors should focus their attention on the importance of this paper (and the main findings) on the use of bismaleimide-based components useful in the construction/building or other industrial applications field.

Conclusions. All of them are quite obvious. Indeed, there are an impressive amount of results. However, the conclusions section needs to improve with selected and highlighted main findings. In conclusion section, it is necessary to more clearly show the novelty of the article and the advantages of the proposed method. Add qualitative or quantitative results of your work. Please try to emphasize your novelty, put some quantifications, and comment on the limitations. This is a very common way to write conclusions for a learned academic journal. The conclusions should highlight the novelty and advance in understanding presented in the work.

-----------------------------------------------------------------------------------------------------------------

The article is interesting but needs to be improved. Authors should carefully study the comments and make improvements to the article step by step. After minor changes can an article be considered for publication in the "Polymers".

Author Response

Dear reviewer,

Re: Revisions requested of Manuscript No.: polymers-1850301.

Thank you very much for giving us an opportunity to revise the manuscript entitled “Preparation and tribological properties of bismaleimide matrix composites reinforced with covalent organic framework coated graphene nanosheets(Manuscript No.: polymers-1850301). Your comments and those of the reviewers are highly insightful and enable us to greatly improve the quality of our manuscript. In the following pages are our point-by-point responses to each of the comments of you. Revisions in the manuscript are shown using yellow highlight [example] for changes.

Comments:

I want to congratulate the authors for the manuscript titled as “Preparation and tribological properties of bismaleimide matrix composites reinforced with covalent organic framework coated graphene nanosheets”. This is an interesting study and the manuscript in its present form is well formatted. I think the results and methods sections are well done with some minor corrections to be made.

  1. Abstract: writing is too generalized. The main theme of this paper is not described in the abstract. Abstract section should be concisely reflected the content and summarize the problem, the method, the results, and the conclusions. Also, please add qualitative or quantitative results of your work. Besides, please avoid starting the sentence with conjunctions such as “and”.

Answer: Many thanks for your attention to our work. In accordance with these suggestions, we have reorganized the abstract section. Some sentences that begin with a conjunction have also been revised.

  1. The introduction section has been written beautifully but need to include recent published papers on bismaleimide-based composites and their production process and overall properties. In this section, more literature papers have to be included to explain the subject better. Some giving citations need to be check such as [7-9] [20-23], as they may not provide the required information in a sentence. In addition, your reference list contains a few papers from Polymers journal. If your work is convenient for this journal’s context, then several references should be included from Polymers journal. (Please only refer to the most relevant and recent studies.)

Answer: Many thanks for your attention to our work. In accordance with these suggestions, we have added recently published papers on bismaleimide-based composites and their production process and overall properties in the introduction section as marked in page 1 line 39. In addition, we have checked the manuscript carefully. Some problems with literature papers have been revised to explain the subject better. Furthermore, we have added a few papers from Polymers journal reference list in the main text.

Zhou et al [2] designed and synthesized a novel eugenol-functionalized cage-like polyhedral oligomeric silsesquioxane (EG-POSS), which exhibits excellent solubility and reactivity with BD-type bismaleimide resin. Thus, 4 wt% incorporated EG-POSS is uniformly dispersed in resin matrix and the resultant hybrid (BDEP-0.04) shows outstanding comprehensive performance, especially on dielectric, mechanical and thermal properties.

[2] Zhang Z.W.; Zhou Y.J.; Cai L.F.; Xuan, L.X.; Wu, X.; Ma, X.Y. Synthesis of eugenol-functionalized polyhedral oligomer silsesquioxane for low-k bismaleimide resin combined with excellent mechanical and thermal properties as well as its composite reinforced by silicon fiber. Chem. Eng. J. 2022, 439: 135740.

[3] Jopen, M.; Degen, P.; Henzler, S.; Grabe, B.; Hiller, W; Weberskirch, R. Polyurea thickened lubricating grease-the effect of degree of polymerization on rheological and tribological properties. Polymers 2022, 14(4), 795.

[4] Chan, J.X.; Wong, J.F.; Petrů, M.; Hassan, A.; Nirmal, U.; Othman, N.; Ahmad Ilyas, R. Effect of nanofillers on tribological properties of polymer nanocomposites: A review on recent development. Polymers 2021, 13(17), 2867.

[46] Lee, P.C.; Kim, S.Y.; Ko, Y.K.; Ha, J.U.; Jeoung, S.K.; Shin, D.; Kim, J.H.; Kim, M. Tribological properties of polyamide 46/graphene nanocomposites. Polymers 2022, 14(6): 1139.

[47] Chan, J.X.; Wong, J.F.; Petrů, M.’; Hassan, A.; Nirmal, U.; Othman, N.; Ilyas, R.A. Effect of nanofillers on tribological properties of polymer nanocomposites: A review on recent development. Polymers 2021, 13(17): 2867.

[50] Chen, X.B.; Lu, S.L.; Sun, C.F.; Song, Z.B.; Jian Kang, J.; Ya Cao, Y., Exploring impacts of hyper-branched polyester surface modification of graphene oxide on the mechanical performances of acrylonitrile-butadiene-styrene. Polymers 2021, 13(16): 2614.

  1. At the last paragraph of the introduction, please clearly show the general outline of the paper and show the importance of the study along with the main aim. Please show the literature gaps demonstrating the presented study fills it.

Answer: Many thanks for your attention to our work, In accordance with these suggestions, we have shown the importance of the study along with the main aim in the last paragraph of the introduction. In addition, we have shown the literature gaps demonstrating the presented study fills it in the last second paragraph of the introduction.

  1. There is an interesting approach and design exists, I just propose to emphasis the practical significance of the presented methodology in several points of article.

Answer: Many thanks for your attention to our work. In accordance with this suggestion, we have emphasized the practical significance of the presented methodology in section 3.3 of the article.

While the addition of the same amount of G/COFs results in a composite material with excellent stability and good friction reduction properties. Meanwhile, the preparation process for G/COFs is straightforward and can be mass produced. So, G/COFs hybrid nanoparticles promising as a new lubricating filler have been applied to the field of friction.

  1. Language used in the manuscript is generally satisfying. However, writers should pay more attention of singular / plural nouns. Also, they should control the spell check/ punctuation of words and sentences. Please check all manuscript for typos and misspellings. Also, please recheck upper/lower case letter and text format for example in abstract line 17 “graphene” is bigger than other words. Please revise such inconsistences.

Answer: Many thanks for your attention to our work. In accordance with these suggestions, we have checked the manuscript carefully. Some problems with spelling, grammar and typos have been revised.

  1. In Fig. 6, some texts are not readable (especially scale bars). Please revise them.

Answer: In accordance with this suggestion, we have added clear scale bars in the Figure 6 and replaced it in the main text.

Figure 6. SEM images of the worn surfaces of neat BMI (a), 0.6 wt% G/BMI (b), 0.6 wt% G/COFs/BMI (c).

  1. The XRD diffraction pattern of Figure 2 should be marked with a standard PDF card. Also, the referred PDF card No. in XRD results should be supplemented in the corresponding text of Fig. 2.

Answer: Many thanks for your attention to our work. The PDF card should be provided to determine the composition of the production. However, we found that the peak at 11.3 corresponds to the 001 crystal plane of graphene oxide during the checking process, not graphite oxide. Here, we have revised the corresponding parts of the manuscript. Meanwhile, we have regretted the PDF cards for graphene oxide and graphene didn't find in database. But to determine their composition, we have cited the relevant reports. The result shown that graphite oxide was transferred to graphene oxide.

Sorry to this careless mistake, and we will pay more attention to it in the future work. Thank you for understanding and support in advance.

[37] Bychko, I.; Abakumov, A.; Didenko, O.; Chen, M.Y.; Tang, J.G.; PStrizhak, P Differences in the structure and functionalities of graphene oxide and reduced graphene oxide obtained from graphite with various degrees of graphitization. J. Phys. Chem. Solids 2022, 164: 110614.

[38] Garcia, J.L.; Miyao, T,; Inukai, J.; V. Tongol, B.J. Graphitic carbon nitride on reduced graphene oxide prepared via semi-closed pyrolysis as electrocatalyst for oxygen reduction reaction. Mater. Chem. Phys. 2022, 288: 126415.

[39]       Liu, Y.; Shin, D.G.; Xu, S.; Kim, C.L.; Kim, D.E. Understanding of the lubrication mechanism of reduced graphene oxide coat-ing via dual in-situ monitoring of the chemical and topographic structural evolution. Carbon. 2021, 173, 941-952.

  1. To put the current work in a broader context, some high-quality literatures should be cited. From my point of view, the topic is surely interesting for the Readers of Polymers as well as the paper is quite well structured. Nevertheless, some results should be better explained avoiding too generic statements (without experimental evidence). In particular, the discussion of the FT-IR results must be checked and mandatory improved in order to increase the quality and readability of this paper. Results are explained without any proper references. Improve the results and discussion and conclusion parts. The results and discussion section should be widened with more focusing point of the findings. And these sentences should be supported with the literature studies. Results and discussion parts are inadequate according to citation and analyze in detail. There should be the importance of the study in detail, comparison results with other approaches in literature, the success of the prediction and computational results.

Answer: Many thanks for your attention to our work. In accordance with this suggestion, we have cited some high-quality literatures in the main text. In addition, the discussion of the FT-IR results is checked and improved. Meanwhile, we have added results that are explained with some proper references. Furthermore, the results and discussion, and conclusion parts have been revised according to your suggestion.

[3] Jopen, M.; Degen, P.; Henzler, S.; Grabe, B.; Hiller, W; Weberskirch, R. Polyurea thickened lubricating grease-the effect of degree of polymerization on rheological and tribological properties. Polymers 2022, 14(4), 795.

[4] Chan, J.X.; Wong, J.F.; Petrů, M.; Hassan, A.; Nirmal, U.; Othman, N.; Ahmad Ilyas, R. Effect of nanofillers on tribological properties of polymer nanocomposites: A review on recent development. Polymers 2021, 13(17), 2867.

[5] Zhang, D.; Li, Z., Klausen, L.H.; Li, Q.; Dong, M.D. Friction behaviors of two-dimensional materials at the nanoscale. Mater. Today Phys. 2022, 27: 100771.

[6] Li. J.f.; Li. J.j; Yi S, Wang, K.Q. Boundary slip of oil molecules at MoS2 homojunctions governing superlubricity. ACS Appl. Mater. Interfaces. 2022, 14(6): 8644-8653.

[7] Serles, P.; Hamidinejad, M.; Demingos, P.G.; Ma, L.; Barri,N.; Taylor, H.; Singh, C.V.; Park,C.B.; Filleter,T. Friction of Ti3C2Tx MXenes. Nano Lett. 2022, 22(8): 3356-3363.

[47] Chan, J.X.; Wong, J.F.; Petrů, M.’; Hassan, A.; Nirmal, U.; Othman, N.; Ilyas, R.A. Effect of nanofillers on tribological properties of polymer nanocomposites: A review on recent development. Polymers 2021, 13(17): 2867.

  1. Furthermore, the Authors should focus their attention on the importance of this paper (and the main findings) on the use of bismaleimide-based components useful in the construction/building or other industrial applications field.

Answer: Many thanks for your attention to our work. In accordance with this suggestion, we have discussed and evaluated the composite materials according to the requirements of construction/building or other industrial applications field.

  1. Conclusions. All of them are quite obvious. Indeed, there are an impressive amount of results. However, the conclusions section needs to improve with selected and highlighted main findings. In conclusion section, it is necessary to more clearly show the novelty of the article and the advantages of the proposed method. Add qualitative or quantitative results of your work. Please try to emphasize your novelty, put some quantifications, and comment on the limitations. This is a very common way to write conclusions for a learned academic journal. The conclusions should highlight the novelty and advance in understanding presented in the work.

Answer: Many thanks for your attention to our work. In accordance with this suggestion, we have reorganized the conclusion section. At the same time, we emphasize the innovation put some quantifications, and comment on the limitations of this work.

We hope that the revisions in the manuscript, supporting information and our accompanying responses will be sufficient to make our manuscript suitable for publication in “polymers”. If you have any queries, please don’t hesitate to contact us at the address below. Thank you and best regards.

Yours Sincerely

Chao Liu

Address: Shaanxi Collaborative Innovation Center of Industrial Auxiliary Chemistry and Technology, Shaanxi University of Science &Technology, Xi’an 710021, China

E-mail: lc1010158@163.com;

Reviewer 2 Report

The authors of the paper which title is “Preparation and tribological properties of bismaleimide matrix composites reinforced with covalent organic framework coated graphene nanosheets” evaluated the synthesis of hybrid lubricating filler and its tribological properties. The paper and the goal is very interesting and the materials and methods and also the discussion part is very well-written, However, authors should consider the following comments before publishing the paper:

-       Please bring the latest studies in the field of polymer-based composite for tribological applications and then highlight the novelty of the paper.

-       For the FTIR analysis, please label all the peaks specially the peaks from 500-1000 cm-1. Moreover, please bring the FTIR analysis of GO as a comparison.

-       For the TEM analysis, please bring the sample preparation in the method part.

In terms of the tribological properties of the composite, please explain the reason of having better performance for the G/COFs/BMI composite. Also compare the results with the literature in a separate paper to have better insight about the results.

Author Response

Dear reviewer,

Re: Revisions requested of Manuscript No.: polymers-1850301.

Thank you very much for giving us an opportunity to revise the manuscript entitled “Preparation and tribological properties of bismaleimide matrix composites reinforced with covalent organic framework coated graphene nanosheets(Manuscript No.: polymers-1850301). Your comments and those of the reviewers are highly insightful and enable us to greatly improve the quality of our manuscript. In the following pages are our point-by-point responses to each of the comments of you. Revisions in the manuscript are shown using yellow highlight [example] for changes.

Comments:

The authors of the paper which title is “Preparation and tribological properties of bismaleimide matrix composites reinforced with covalent organic framework coated graphene nanosheets” evaluated the synthesis of hybrid lubricating filler and its tribological properties. The paper and the goal is very interesting and the materials and methods and also the discussion part is very well-written, However, authors should consider the following comments before publishing the paper.

  1. Please bring the latest studies in the field of polymer-based composite for tribological applications and then highlight the novelty of the paper.

Answer: Many thanks for your attention to our work. In accordance with this suggestion, we added the latest studies in the field of polymer-based composite for tribological applications and then highlight the novelty of the paper in the Introduction part.

  1. For the FTIR analysis, please label all the peaks specially the peaks from 500-1000 cm-1. Moreover, please bring the FTIR analysis of GO as a comparison.

Answer: Many thanks for your attention to our work. In accordance with this suggestion, we have discussed this phenomenon in FTIR analysis.

The peaks at 824 cm−1 and 708 cm−1 belong to the characteristics of substitution peaks on benzene rings [27, 28, 36].

Graphene oxide exhibits three clear absorption peaks at 1062, 1315, and 3600 cm−1, which belong to C-O-C stretching vibrations, C-H bending vibration, and -OH stretching vibrations, respectively [16, 34].

[16] Liu, C.; Li, X.; Lin, Y.; Xue, X.; Yuan, Q.Y.; Zhang, W.B.; Bao, Y.; Ma, J.Z. Tribological properties of bismaleimide-based self-lubricating composite enhanced by MoS2 quantum dots/graphene hybrid. Compos. Commun. 2021, 28, 100922.

[27] Shinde, D.B.; Aiyappa, H.B.; Bhadra, M.; Biswal, B.P.; Wadge, P.; Kandambeth, S.; Garai, B.; Kundu, T.; Kurungot, S.; Banerjee, R. A mechanochemically synthesized covalent organic framework as a proton-conducting solid electrolyte. J. Mater. Chem. 2016, 4(7), 2682-2690.

[28] Li, W.; Zhang, X.Y.; Zhang, C.Y.; Yu, M.Y.; Ren, J.F.; Wang, W.; Chen, S.G. Exploring the corrosion resistance of epoxy coat-ed steel by integrating mechanochemical synthesized 2D covalent organic framework. Prog. Org. Coat. 2021, 157, 106299.

[34] Liu, C.; Yin, Q.; Zhang, W.B.; Bao, Y.; Li, P.P; Hao, L.F. Tribological properties of graphene-modified with ionic liquids and carbon quantum dots/bismaleimide composites. Carbon. 2021, 183, 504-514.

[36]Yin, C.C.; Fang, S.Y.; Shi, X.S.; Zhang, Z.; Wang, Y. Pressure-modulated synthesis of self-repairing covalent organic frame-works (COFs) for high-flux nanofiltration. J. Membrane. Sci. 2021, 618, 118727.

  1. For the TEM analysis, please bring the sample preparation in the method part.

Answer: In accordance with this suggestion, we added the TEM sample preparation in the method part.

A small amount of the sample was dispersed into ethanol and sonicated to make it uniformly dispersed. The sample was dropped onto a 230-mesh carbon support film to obtain a transmission electron microscope sample.

  1. In terms of the tribological properties of the composite, please explain the reason of having better performance for the G/COFs/BMI composite. Also compare the results with the literature in a separate paper to have better insight about the results.

Answer: Many thanks for your attention to our work. In according with your suggestion, we have explained the reasons of having better performance for the G/COFs/BMI composite. The friction coefficient reduction of G/COFs for BMI is significantly better than that of similar literatures [16, 45].

The outstanding tribological properties of the G/COFs/BMI composites are mainly attributed to two aspects. Firstly, graphene has excellent reinforcement efficiency and self-lubricity, so the introduction of graphene into the polymer matrix can effectively improve the mechanical strength of the composite, thereby improving the wear resistance of the composite (See Figure S1). Secondly, the deposition of COFs can not only effectively prevent the aggregation of graphene nanosheets, but also significantly improve the compatibility and interfacial bond between graphene and polymer matrix. Therefore, the G/COFs/BMI can form a more uniform and continuous high-quality self-lubricating transfer film during the friction process, so as to effectively improve the tribological properties of the composites.

We hope that the revisions in the manuscript, supporting information and our accompanying responses will be sufficient to make our manuscript suitable for publication in “polymers”. If you have any queries, please don’t hesitate to contact us at the address below. Thank you and best regards.

Yours Sincerely

Chao Liu

Address: Shaanxi Collaborative Innovation Center of Industrial Auxiliary Chemistry and Technology, Shaanxi University of Science &Technology, Xi’an 710021, China

E-mail: lc1010158@163.com;

Reviewer 3 Report

The presented paper contains interesting new data on obtaining G/COFs/BMI composite with a complex of improved characteristics. The presented work is of scientific and applied interest and can be accepted for publication.

The reviewer has a number of comments/recommendations on the content and design of the manuscript.

1. For all reagents (section 2.1), except for manufacturing companies, the characteristics of reagents (content of main components and qualification) should be specified.

2. When stating the brands of instruments, consistency must also be maintained (if the country of manufacture is indicated, then everywhere).

3. it is desirable to give in more detail in the text the modes of obtaining G/COFs/BMI composite (temperature, time, etc.).

4. Zoom in on the SEM images (Fig.6).

5. Were such parameters of composite material as elongation at break, tensile strength, Young's modulus investigated? If such data are available, they should be given. 

Author Response

Dear reviewer,

Re: Revisions requested of Manuscript No.: polymers-1850301.

Thank you very much for giving us an opportunity to revise the manuscript entitled “Preparation and tribological properties of bismaleimide matrix composites reinforced with covalent organic framework coated graphene nanosheets(Manuscript No.: polymers-1850301). Your comments and those of the reviewers are highly insightful and enable us to greatly improve the quality of our manuscript. In the following pages are our point-by-point responses to each of the comments of you. Revisions in the manuscript are shown using yellow highlight [example] for changes.

Comments:

The presented paper contains interesting new data on obtaining G/COFs/BMI composite with a complex of improved characteristics. The presented work is of scientific and applied interest and can be accepted for publication. The reviewer has a number of comments/recommendations on the content and design of the manuscript.

  1. For all reagents (section 2.1), except for manufacturing companies, the characteristics of reagents (content of main components and qualification) should be specified.

Answer: In accordance with this suggestion, we have added the characteristics of reagents (content of main components and qualification) in section 2.1.

  1. When stating the brands of instruments, consistency must also be maintained (if the country of manufacture is indicated, then everywhere).

Answer: In accordance with this suggestion, we have checked section 2.5 carefully. We have revised the brands of the instrument's information.

  1. It is desirable to give in more detail in the text the modes of obtaining G/COFs/BMI composite (temperature, time, etc.).

Answer: Many thanks for your attention to our work. In accordance with this suggestion, we have provided detailed modes of obtaining G/COFs/BMI composite in section 2.4.

The G/COFs/BMI composites were prepared by mixing the pre-weighed quantities of G/COFs, DBA, and BDM (DBA and BDM with a mass ratio of 3:4). Then the mixture was heated to 135°C till the mixture totally melted and the G/COFs dispersed uniformly. Finally, the mixture was cured following the schedule of 150 °C/2 h + 180 °C/2 h + 220 °C/4 h. The post-curing process was 250 °C/4 h.

  1. Zoom in on the SEM images (Fig.6).

Answer: In accordance with this suggestion, we have zoomed on the SEM images resolution (Fig.6).

Figure 6. SEM images of the worn surfaces of neat BMI (a), 0.6 wt% G/BMI (b), 0.6 wt% G/COFs/BMI (c)

  1. Were such parameters of composite material as elongation at break, tensile strength, Young's modulus investigated? If such data are available, they should be given.

Answer: Many thanks for your attention to our work. In accordance with this suggestion, the flexural strength and impact strength of G/COFs/BMI composites was studied and discussed in the supporting information Figure S1. Meanwhile, we have added the results to section 3.3.

When served in applications with high load and sliding velocity, it is of great importance for the composites to have outstanding flexural strength and impact strength to resist deformation and bending failure. In this study, the dependency of the flexural strength and impact strength of the G/COFs/BMI composites on the content of the G/COFs is shown in Figure S1.

It can be seen that the composites containing the G/COFs exhibit higher strength values than those of the G at almost all the filler content (See Figure S1 a, b). As shown that the flexural strength and impact strength of composites fall at first, then rise as the number of fillers increases. When the filler addition is 0.6 wt%, the flexural strength and impact strength of G/COFs/BMI composite reach the maxima with 151 MPa and 15.6 kJ/m2. Compared to neat BMI (120 MPa and 12.3 kJ/m2), it is improved by 25.8% and 26.8%, respectively. This phenomenon is attributed to the unique sheets structure of hybrid graphene and the interfacial adhesion between G/COFs and BMI matrix, which can greatly improve the flexural strength and impact strength of BMI resin[1]. However, when the addition amount is more than 0.6 wt%, the flexural strength and impact strength of G/COFs/BMI composites decrease. But they are still higher than those of neat BMI. It may be because excessive fillers can not be well dispersed in the BMI matrix and agglomerate to cluster [2]. Consequently, the advantages of the fillers can not be fully utilized and the flexural strength and impact strength of the composites decreases with the uneven distribution of the fillers in the BMI matrix [3].

Figure S1. Relationship between mechanical properties and fillers content: flexural strength (a) and impact strength (b) of the composites, respectively.

[1] Chan, J.X.; Wong, J.F.; Petrů, M.; Hassan, A.; Nirmal, U.; Othman, N.; Ahmad Ilyas, R. Effect of nanofillers on tribological properties of polymer nanocomposites: A review on recent development. Polymers 2021, 13(17), 2867.

[2] Liu, C.; Dong, Y.F.; Lin, Y.; Yan, H.X.; Zhang, W.B.; Yan, B.; Ma, J.Z. Enhanced mechanical and tribological properties of gra-phene/bismaleimide composites by using reduced graphene oxide with non-covalent functionalization. Compos. Part B Eng. 2019, 165, 491-499.

[3] Zhang, Y.B; Yan H.X; Xu, P.L.; Guo, L.L; Yang, K.M.; Rui Liu, R.; Feng, W.X. A novel POSS-containing polyimide: Synthesis and its composite coating with graphene-like MoS2 for outstanding tribological performance. Prog. Org. Coat. 2021, 151: 106013.

We hope that the revisions in the manuscript, supporting information and our accompanying responses will be sufficient to make our manuscript suitable for publication in “polymers”. If you have any queries, please don’t hesitate to contact us at the address below. Thank you and best regards.

Yours Sincerely

Chao Liu
